# LCA of Wood Waste Management Systems: Guiding Proposal for the Standardization of Studies Based on a Critical Review

Giusilene Costa de Souza Pinho * and João Luiz Calmon

Department of Environmental Engineering, Federal University of Espírito Santo, Fernando Ferrari Avenue, 514, Goiabeiras, Vitória 29075-910, Brazil
* Correspondence: giusilenecosta@gmail.com

**Abstract:** In environmental management, there are many opportunities to improve wood waste (WW) management practices. Life Cycle Assessment (LCA), according to ISO 14040, is a tool used to assess the environmental impacts related to a product throughout its life cycle. Thus, this article aims to propose guidelines for the creation of future LCAs of wood waste management systems in a consistent and standardized way based on the deficiencies and examples found in the studies that comprise the extensive bibliographic review of this research. During the selection of studies, the methodology termed *Methodi Ordinatio* was used, which considers the three most relevant points to qualify a scientific study: the impact factor; the year of publication; and the number of citations. Fifty (50) articles were identified to create a general map of the literature relevant to the topic under study. We carried out a critical review that highlights the lack of standardization and clarity of the research in this area. For example, in relation to the total number of studies analyzed, 67% did not clarify the type of analysis (attributional or consequential). Several recommendations and perspectives within the LCA of WW management were highlighted, such as the need to analyze impact categories other than climate change and to include economic and social analyses in new studies. In order to leverage all these research opportunities, it is important that LCA practitioners adopt global standardization. In future research, the guiding proposal presented in this study can improve the comparison between scenarios and the consistency of results.

**Keywords:** life cycle assessment; LCA; wood waste; critical review; guiding proposal

## 1. Introduction

The economic value of products is the main focus of a market-oriented economy, but this perspective neglects the depletion of natural resources and the increasing amount of waste. In the short term, if the recycling of natural resources and waste management do not receive their due attention, many of our planet's reserves will soon be extinguished [1].

In this context, wood waste is still underused. In Brazil, approximately 30 million tons of wood waste is generated annually, with the timber industry accounting for 91% of the total, followed by waste from the urban environment and civil construction with 6% and 3%, respectively [2].

Wood waste can be used for the manufacturing of materials and energy production [3]. These residues can be applied as raw materials in the pulp industry, in the production of panels, and as oven fuels in the generation of thermal or electric energy, among others. In 2019, Brazil's planted forest industry reported that most of its products (67%) were used for power generation, while fractions of only 12% were allocated as raw material in other industries and those of 21% were discarded in other ways [4]. These data need attention; the disposal of wood wastes in landfills is not environmentally sustainable due to greenhouse gas emissions and competition with other uses of the land [5].

Virgin raw materials can be replaced by wood waste, thereby reducing environmental impacts and the costs for extraction, transport, and disposal, such as incineration or transport to a landfill. Through the recycling of timber residue, lower environmental burdens can

be achieved by reducing the materials, water, and energy used in the production processes when compared with the use of virgin raw materials [5]. Among the many potential wood waste byproducts, the production of wood-derived panels can accommodate a significant number of such byproducts. MDP (Medium-Density Particleboard), for example, offers great compatibility and can receive up to 100% of residues, depending on the country [3,6]. In 2020, the level of the consumption of panels was 6.9 million m$^3$ in Brazil, of which 2.8 million was related to MDP [4].

Many studies have considered woody biomass and wood products to be "carbon neutral," but the life cycle literature has questioned this assertion of absolute carbon neutrality [7–9]. The studies on the life cycle assessment (LCA) of the forestry sector primarily discuss the impacts on global warming potential (GWP), since the resources derived from wood mainly comprise biogenic carbon, which facilitates the mitigation of climate change by moving fossil fuels through the recovery of energy or by storing carbon within wood products for a long period [3].

Since wood waste is considered a recyclable and renewable resource, environmentally friendly solutions for transforming it into value-added products are required for its recycling. Consequently, the residue can be used in production systems, thereby reducing the discarded volumes and the demand for virgin sources of wood. These alternatives for the disposal of wood waste can leverage the Circular Economy (CE) within the forestry sector [10–12]. The Circular Economy, however, transcends waste management; it is a restorative and regenerative industrial model by intention and design. Thus, the CE replaces the concept of the "end of life" with restoration, promotes the use of renewable energy, stops the use of toxic chemicals that impair reuse, and aims to eliminate waste with the design of materials, products, and systems [13].

Studies on Life Cycle Evaluation have highlighted in their analyses the principles of the CE [11,14–17]. According to ISO 14040 (ISO, 2006a) [18], the Life Cycle Assessment methodology is a quantitative method of analyzing the environmental impacts that govern a product's life cycle from the extraction of raw materials to the final disposal of materials after use. Based on this systemic view, environmental loads can be identified and avoided. In this context, LCA can be used as a tool to leverage the CE [10].

However, putting these concepts into practice is still a great challenge. For example, wood waste is still insignificantly used in most countries due to the lack of efficiency in classifying the types of this waste, but there are possibilities for its greater use through improvements in sorting technologies [3,19]. Taskhiri et al. (2019) [20] highlighted that chemical contamination is generally much higher for unselected wood residues, showing that adequate selective collection, classification, and the correct handling of waste can improve and promote its insertion in new cycles with more efficient recycling practices.

Public policies have been encouraging the use of the life cycle assessment (LCA) methodology to identify more efficient residue management strategies [21]. European legislation is an important example and, in Brazil, Law No. 12.305/2010 [22], which deals with the National Policy on Solid Waste Management, includes concepts of solid waste management, life cycle management, and shared responsibility, which promotes the use of LCA as a management tool.

Regarding environmental management, there are many opportunities to improve the management practices applied to wood waste, but the interpretation and comparison of different LCA studies focusing on this area is still a difficult task.

LCA is the main method for assessing the environmental impacts for wood waste management systems. Comparative LCA was used to make safe decisions with respect to different management scenarios for wood waste originating from civil construction in Hong Kong [14]. The method was used to explore the environmental performance of bioenergy production in an Alpine area of northern Italy [16]. The impact of climate change and uncertainties related to emissions from wood-based energy waste in the UK were examined using LCA [8]. LCA was also used to evaluate the emission of greenhouse gases in the life

cycle of bioconcrete production with the addition of wood chips in Brazil [17]. Furthermore, in the literature that focuses on wood cascades, the use of LCA is broad [3,19,20].

Thoneman and Schurmann (2018) [23] conducted a systematic review of the literature—incorporating 15 publications—to identify the environmental impacts of the cascading use of wood according to their respective measurement methods. The authors observed that LCA is the most used evaluation method and that different functional units have been adopted to evaluate the environmental impacts of the cascading use of wood systems. Regarding the type LCA conducted, the authors suggest that attributional LCA studies should be expanded into a consequential LCA. They also concluded that data collection and the modeling of a cascading system should be more consistent and realistic.

The study conducted by Thoneman and Schurmann (2018) [23] focused on the environmental performance of the cascading use of wood; to date, no other study within the surveyed databases has reviewed the application of the LCA methodology to evaluate wood waste management systems more broadly, in relation to the other objectives addressed in the literature. Therefore, there is a scientific gap that this research aims to bring to light, contributing an extensive review and proposing guidelines for the standardization of future studies in the area.

In view of the above, the objective of this article is to formulate a guideline proposal for the elaboration of future LCAs of wood waste management systems in a consistent and standardized manner and based on the deficiencies and good examples found in the studies that comprise the bibliographic portfolio of this review. The scope of this review also allows us to determine the methodological aspects of LCA used in this field of study; the wood waste by-products adopted in the research; the origin of wood waste; the main contributors to the impact categories; promising solutions; approaches to the circular economy; and the main perspectives for future studies.

## 2. Methods

The *Methodi Ordinatio* was the methodology used to select studies for the bibliographic portfolio. This methodology differs from others used in systematic reviews by its use of InOrdinatio, which is an index that allows for the classification of studies according to scientific relevance [24]. The equation considers the three most important points to qualify an article: its impact factor, year of publication, and number of citations [25].

The Scopus and Web of Science databases were used in this study due to their compatibility with the Bibliometrix tool, which was programmed in the R software [26]. Initially, a preliminary search was carried out on the databases to evaluate and test the adherence to the selected keywords and to find other terms in line with the research. Thus, to search the relevant literature, the keywords "life cycle assessment" or "LCA" and "wood waste" or "wood residue*" were selected. Additionally, the words "cascad*" and "wood" or "timber" were added after searching for other terms correlated with the topic.

To ensure relevance, the filters for "article" and "review," and the period from 2011 to 2021 were used. The chosen period had to be broader in order to include the more traditional articles in the portfolio. The studies, however, were only included if they presented a positive inOrdinatio index resulting from the applied equation [24].

First, 107 articles were selected after the elimination of duplicates. Then, after reading the titles, abstracts, and keywords, 54 articles were chosen. Subsequently, a reading of the studies that still held doubts was performed. We discarded studies that only mentioned LCA but did not use this methodology and studies that mentioned wood waste but conducted LCA for other purposes, among others. Finally, 50 articles (01 Revision and 49 original articles) were classified to make up the portfolio of this study, of which all had a positive inOrdinatio index (Figure 1).

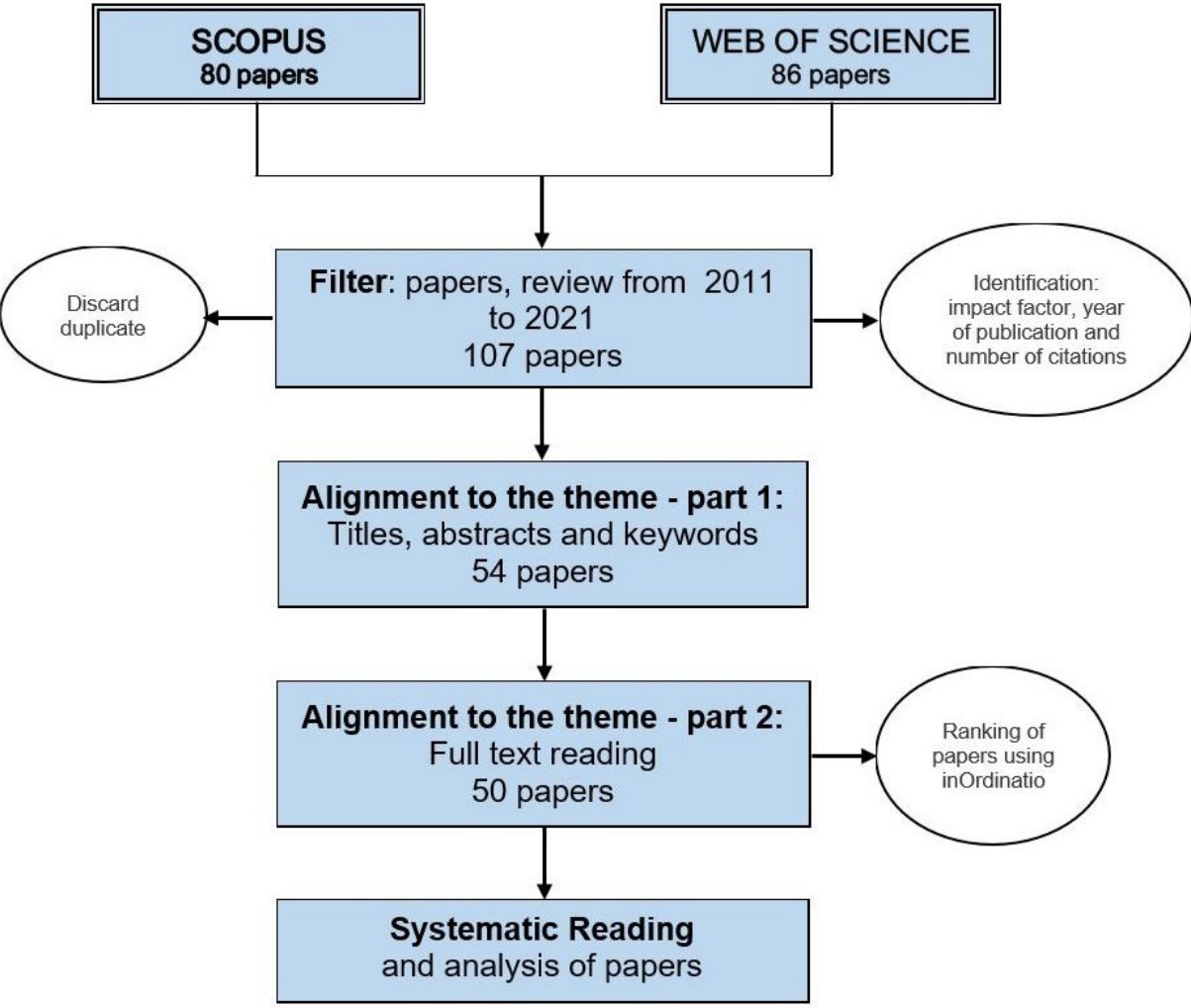

**Figure 1.** Methodology used for selection of LCA research for the management of timber residue.

The "Science Mapping" spreadsheet created in Microsoft Excel 2016 by the Center for Organizational Performance of the Federal University of Espírito Santo was used to import and treat the data extracted from R (or R Studio) using the Bibliometrix library. Item 3.1 shows the main results.

After the creation of the portfolio, the 49 original articles were transferred to a new spreadsheet in Excel; then, they were thoroughly read so as to be analyzed quantitatively and qualitatively. In addition to the methodological aspects of LCA, the articles were characterized in relation to the types of objectives, the most important results concerning the scenarios studied, and the main impacting contributions. Other points were also recorded, such as whether social and economic aspects were contemplated, the origin of the timber residue, and whether the articles highlighted points in line with the concepts of a Circular Economy.

## 3. Results and Discussion

### 3.1. Bibliometric Results

Figure 2 shows the evolution of research over a decade. The year with the most published studies was 2019, and the journal with the highest incidence of articles in the studied area was the Journal of Cleaner Production (32%), which published four times as many studies than the second-place journal (Figure 3).

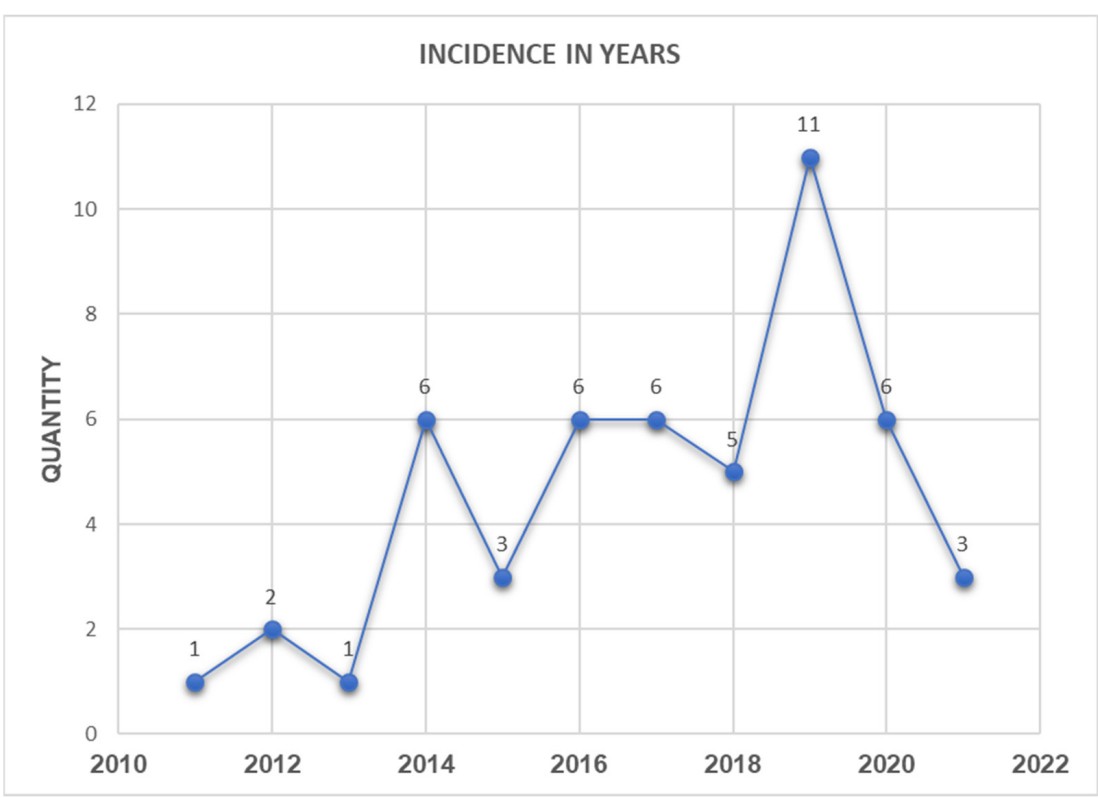

**Figure 2.** Temporal evolution of publications related to the LCA for timber waste management, from 2011 to 2021.

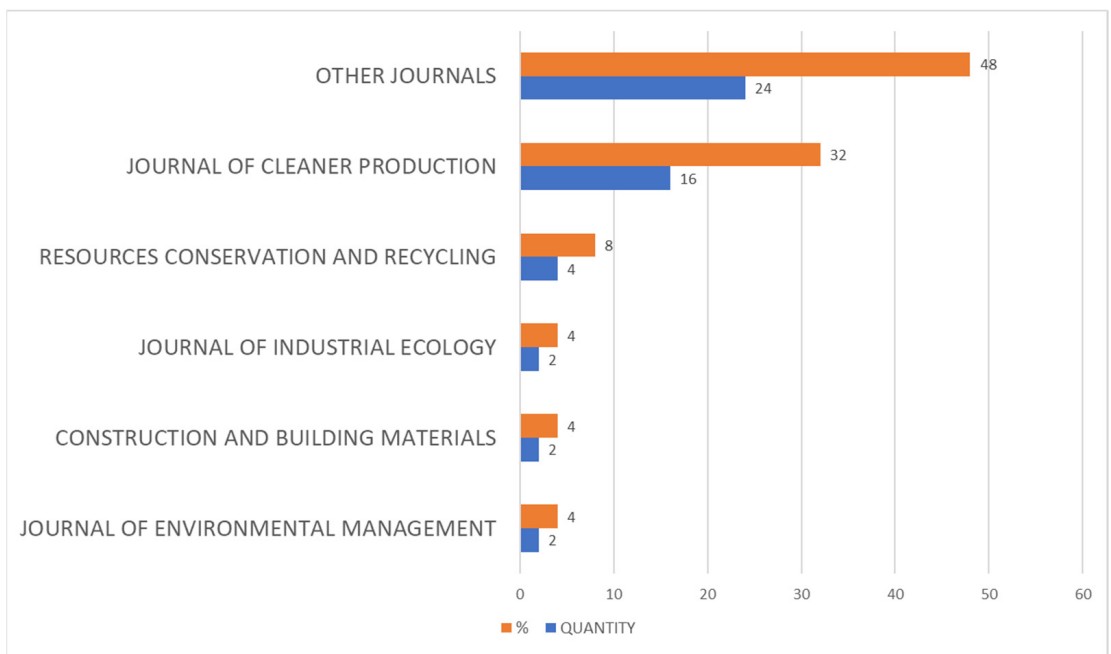

**Figure 3.** Incidence by journals of publications related to the LCA of wood waste management from 2011 to 2021. Note: Journals that have only 1 publication were represented under the item "other journals".

Germany (8), Canada (6), Italy (6), and the United States (5) were the countries with the highest number of publications in the field (Figure 4). There are many countries that are still underrepresented in relation to the number of published papers.

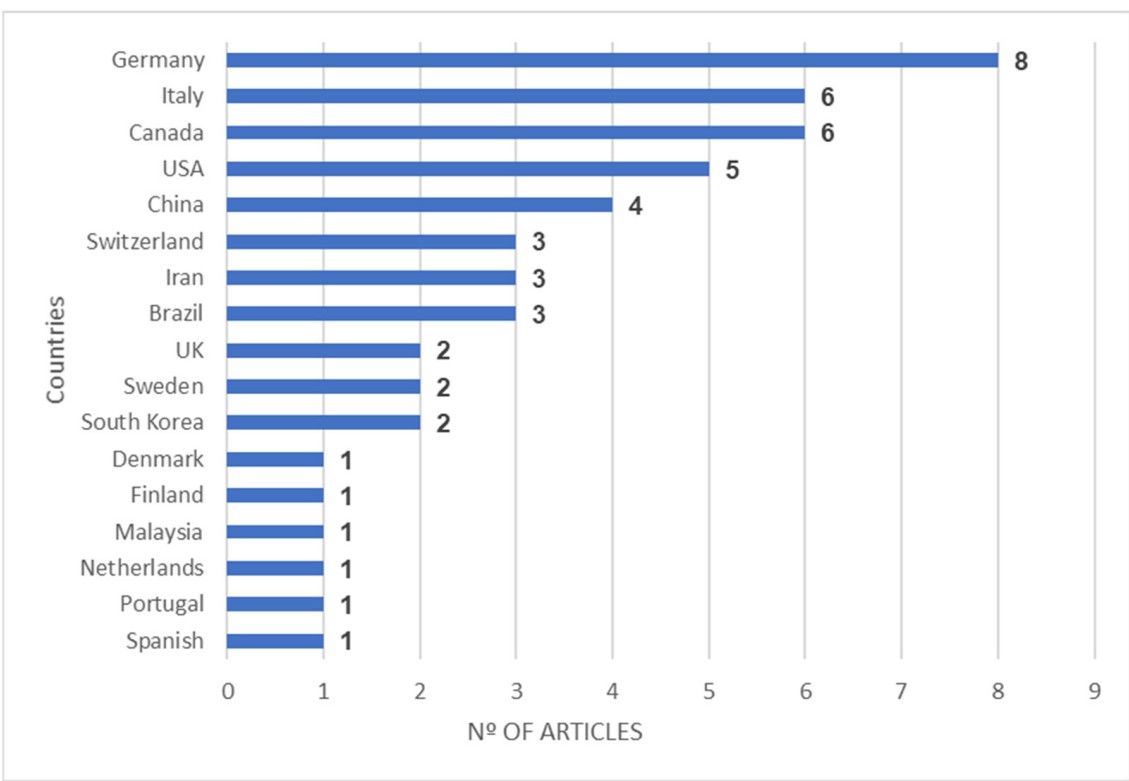

**Figure 4.** Quantitative analysis of publications of research related to LCA for wood waste management from 2011 to 2021 by countries of origin of the studies and considering the main author.

Figure 5 shows that Europe has a remarkable number of countries that have already published on the subject. In North America, the United States, and Canada have a total of 11 studies. While in South America, only Brazil appears, with three studies. Asia conducted 10 studies between 2011 and 2021. The other continents have not yet published studies on the subject, which shows potential for further research.

### 3.2. Systematic Bibliographic Review

The 50 articles (49 originals and 1 review) selected to make up the bibliographic portfolio of this study were mapped and analyzed in detail. In the original articles, the methodological aspects of LCA were examined, for which ISO 14040 (2006a) [18] and ISO 14044 (2006b) were used as references [27]. Thus, the analyses presented from item 3.2.1 follow the structure of the LCA phases, encompassing the definition of the objective and scope, the analysis of the life cycle inventory (LCI), the life cycle impact assessment (LCIA), and the interpretation phase.

#### 3.2.1. Definition of Objective and Scope

The objective of the studies was consistently presented by 100% of the 49 articles. In total, 49% of studies applied LCA to evaluate energy production from wood waste (EPWW), 17% analyzed wood waste management strategies (WWMS), and 14% constituted studies on the use of residue within the cascading use of wood (CUWW). Products manufactured with virgin material versus recycled material (VP × RP) were present in 16% of the studies, while only 4% studied the manufacture of any product with the incorporation of wood waste [28,29]. Energy production systems based on the use of wood waste were the main topic of research analysis, reflecting the incentivization of many governments with respect to renewable alternatives.

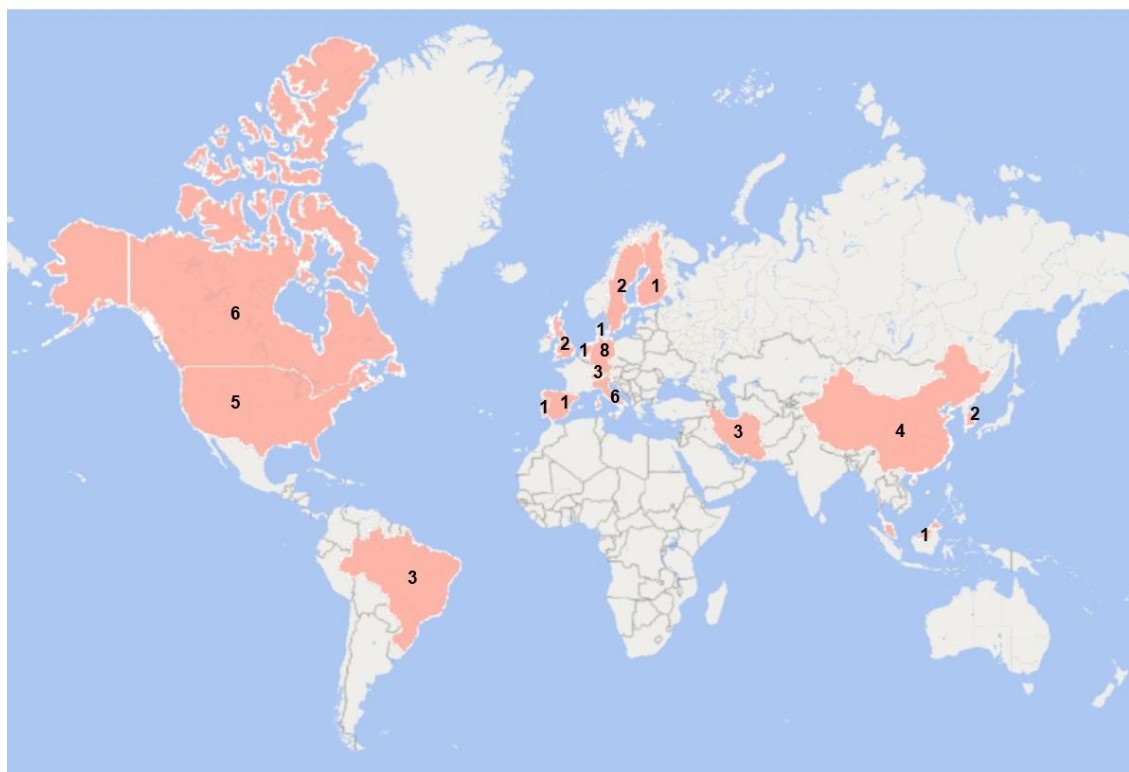

**Figure 5.** Geographical and quantitative distribution of publications of research related to LCA for the management of wood waste from 2011 to 2021 classified by countries of origin of the studies and considering the main author.

The use of wood waste for energy production and product manufacture can contribute to the replacement of fossil fuels and carbon storage, respectively [17]. Figure 6 shows that heat and electricity, both consisting of 57% of the studies, were the most studied applications. In relation to wood products, MDP was the most investigated (31%), followed by pellet fuels (14%), MDF (10%), and OSB (8%). The number of studies that focused on MDP is justified by the product's capacity with respect to incorporating waste material while maintaining mechanical properties similar to the particleboards produced with virgin wood. However, other materials with great potential, such as soundproof boards, hydraulic tiles, and ceramic materials, are still little explored or have not yet been an object of study, which opens the field for further studies regarding LCA.

About 82% of the studies presented their scope clearly, establishing the analyzed systems, the functional unit, and the limits of the system. The remaining 18% were classified as deficient due to either omitting the functional unit used or failing to present a definition of the boundaries of the studied system.

One ton of wood waste was the functional unit that was present in the different types of objectives, being adopted by 14% of the 49 studies. Another 14% of the studies adopted 1 m$^3$, and this unit was the most used in studies that had VP × RP or RP as an objective. The unit most used in research, which evaluated the amount of energy supplied by a given system, was 1 kWh, representing 8%. Units related to a complete product, such as a cabinet, were found in 6% of the studies [6,30,31]. Units related to annual production were used in 4% of the studies [20,32]. In addition, 4% of the studies adopted m$^2$ as a functional unit [33,34]. Among the studies, 8% did not report the functional unit used in their research. The remaining studies (42%) used varied units, such as the distance of 1 km traveled [35].

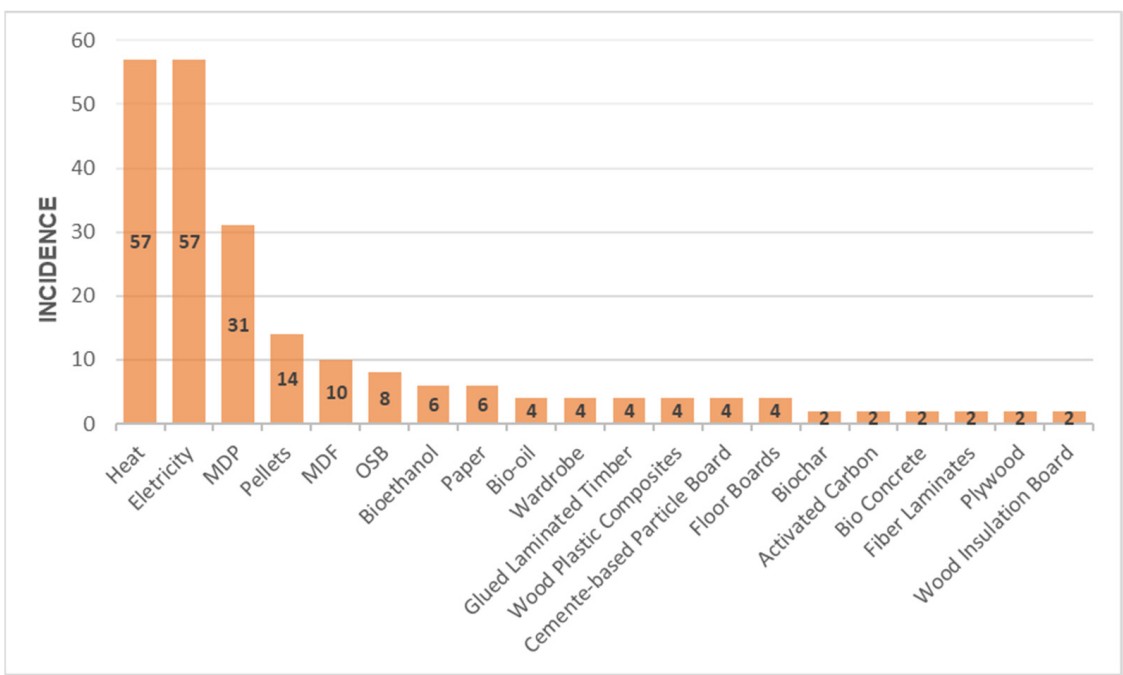

**Figure 6.** Application of wood waste in LCA studies from 49 selected articles. Note: Most of the studies analyze more than one application for the residue.

Most studies (94%) used untreated wood waste. Few studies (6%) used residues with contamination rates [3,8,20]. Regarding the sources of origin, 43% used industrial waste; 29% used wood waste from sawmills; and 18% used construction and demolition waste. This was followed by the analysis of forest residue, accounting for 16%; post-consumer wood waste, at 14%; and agricultural residue, at 12%. In addition, only 6% highlighted wood waste from a furniture factory as the source of origin. The authors treat industrial wood residues as a single residue; in this regard, some studies possibly included wood residue from furniture manufacture. Few studies explicitly stated, for example, whether the residues originated from the wood paneling industries [17]. Figure 7 shows an overview of the origins of the residues considered in the studies.

About 73% of the studies reported the adopted waste treatment. In this group, most studies used mechanical treatments such as grinding, sieving, and drying. Only one study considered thermo-hydrolytic disintegration [20], a method in which pressure and steam are used to separate resin from residual wood. Since these processes are relevant in terms of determining possible environmental impacts, they should be well-defined within the study. However, 27% of the studies did not report the type of treatment given to the residue.

Due to the different approaches to research, the consideration of the boundaries of the system did not always occur in the same way. For 45% of the studies, the extension of the cradle-to-grave process was chosen. Depending on the study, however, the "cradle" could begin during the cutting and harvesting of wood in the forest, whereas other studies considered it to be from the wood residue's use in collection and processing centers. For the product-manufacturing sector, the "grave" phase encompasses the process of incineration or dumping in a landfill, whereas in the energy sector, it is the phase of conversion into electricity or heat. About 31% of the studies considered the limit of the cradle-to-gate of the industry. Analyses that focused solely on the gate-to-gate process represent 14% of the works. However, 8% of the studies did not specify the boundary adopted, and none of the 49 studies analyzed considered the return to the cradle (cradle-to-cradle). The study by Bello et al. (2020) [35] made a differentiated consideration by adopting the limits from biomass growth to the final use of the biofuel in a passenger vehicle (cradle-to-wheel).

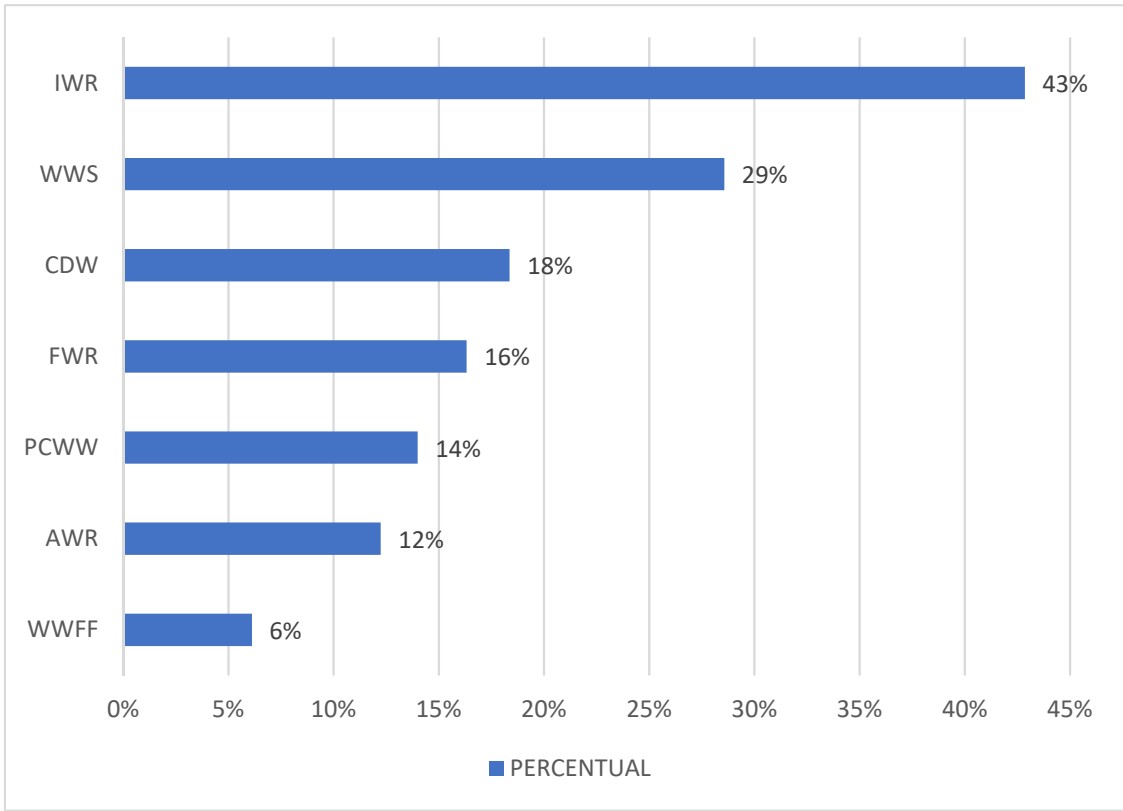

**Figure 7.** Origin of wood waste considered in LCA studies, from 49 selected studies. Note: WWS—Wood waste from sawmills; PCWW—Post-consumer wood waste; WWFF—Wood waste from furniture factory; IWR—Industrial wood residues; FWR—Forest wood residues; CDW—Construction and demolition waste; AWR—Agricultural wood residues. The nomenclatures have been presented as they appeared in the surveys and some authors considered certain wood wastes to originate from more than one source.

Regarding the use of wood ash, it was found that only 10% of the 49 studies included it in their analyses. Corona et al. (2020) [36] considered the use of wood ash to replace limestone in the production of asphaltic bitumen pavement, with 95% of the ash destined for concrete. The rest of the bottom ash and fly ash were directed to landfills. Another application was in agriculture as a substitute for potassium fertilizer [37,38]. The other authors considered the transportation of wood ash to landfills.

Most studies, around 67%, did not report whether the analysis was conducted as an attributional or consequential LCA study. The attributional approach seeks to generate information about which part of the global burden may be linked to a product, and the consequential approach is used to provide information about the environmental burdens that arise, directly or indirectly, from a decision, for example, an increase in product demand [39]. In the studies that explained the type of analysis conducted, 27% used the attributional LCA, while only 6% used consequential LCA [3,33,36,40].

### 3.2.2. Life Cycle Inventory Analysis

About 73% of the studies highlighted the use of primary collected data and 27% did not report their use. The secondary data employed mainly stemmed from the Ecoinvent or Gabi databases and from the relevant literature. Other databases were also used, such as the National Life Cycle Inventories Database (Banco Nacional de Inventários do Ciclo de Vida—SICV Brazil) [11]; the United States Life Cycle Inventory Database (USLCI) [15,38,41,42]; the European Life Cycle Inventory Database (ELCD) [3]; and GHGenius [43–45].

### 3.2.3. Assessment and Interpretation of Life-Cycle Impact

The most used software was SimaPro, which was present in 47% of the surveys; 12% used Gabi, 4% used Umberto, and 6% used other software such as Easetech [3] and GEMIS [46]. In this group, 76% indicated the version of the software used, and 31% of the studies did not specify the software adopted, which might have influenced the replicability of the results found.

Figure 8 shows the LCIA methods adopted in the LCA research applied to the use of wood waste, and it is possible to verify that the ReCiPe (18%) and IPCC (18%) methods were the most used within the midpoint categories (Figure 8a). While IPCC enables analysis of only one category, ReCiPe enables extensive analysis by uniting midpoint and endpoint approaches into a single structure. ReCiPe was created from Eco-indicator 99 and CML 2000, with global applications for the categories of the impact of climate change, the destruction of the ozone layer, and the consumption of resources. In the group of articles that used endpoint categories (Figure 8b), ReCiPe also stood out, and was adopted by 14% of the 49 studies reviewed. Within the midpoint approach, CML (16%) and TRACI (12%) stand out. The first has a more global scope of application while the second is based on the conditions of the United States.

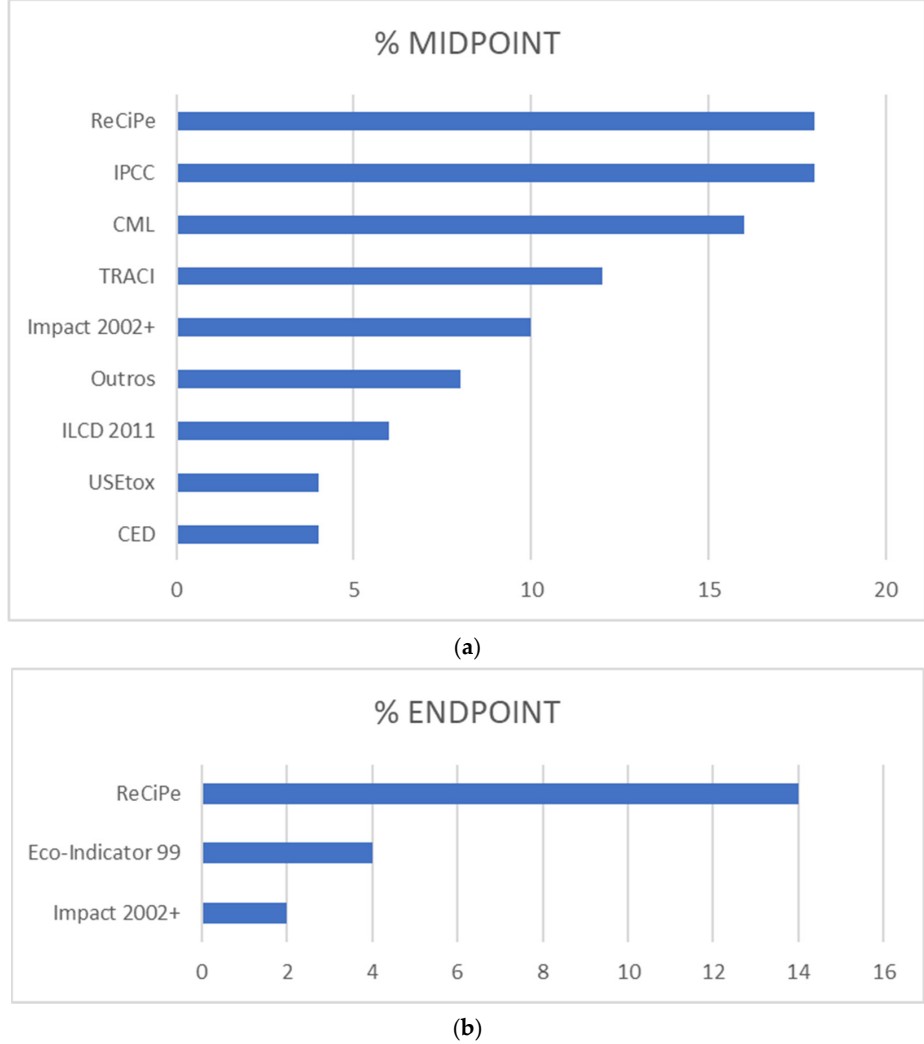

**Figure 8.** LCIA methodologies applied in LCA studies to evaluate the environmental performance of wood waste use in the 49 selected articles. (**a**) studies that adopted the midpoint category. (**b**) studies that adopted the endpoint category. Note: Some studies used more than one methodology. In addition, a single type of methodology was also found in different versions among the selected studies.

Midpoint categories were used in 70% of the studies, while endpoints were adopted by 2% [30] and 16% used both categories. However, about 12% of the analyzed articles did not explain the method used to reach the result of the indicator presented. Figure 9 shows that 86% of the studies analyzed the category of "global warming." Subsequently, "acidification," representing 53%; "eutrophication," representing 47%; and "human toxicity," representing 43% also showed relevance. Most of the research in the forestry sector analyzed impacts considering the indicator for Global Warming Potential (GWP), since the resources of wood are basically constituted by biogenic carbon [3,17].

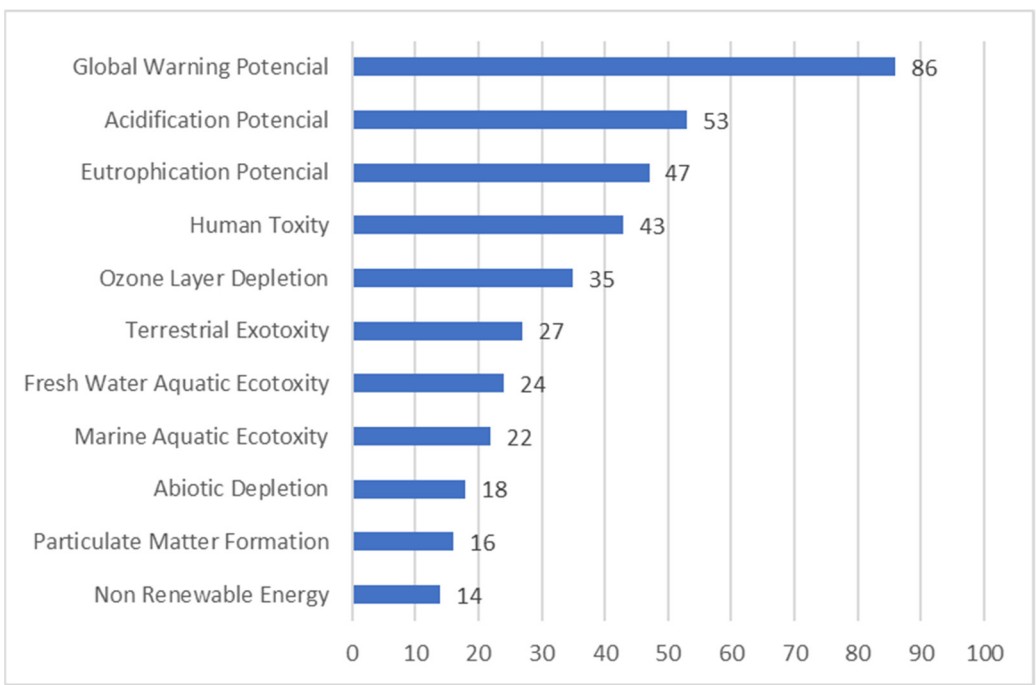

**Figure 9.** Impact categories most applied in LCA studies to evaluate the environmental performance of wood waste use in the 49 selected articles. Note: Other categories were also studied, with membership lower than 14%.

About 47% of the studies performed a sensitivity analysis, and only 18% presented an uncertainty analysis. These items are very important for the greater credibility of studies related to LCA and should be included in such studies as a resource to measure the consistency of the results.

*3.3. The Results Achieved and the Sustainable Development Goals (SDGs)*

The results achieved in this study were interpreted in light of the Sustainable Development Goals defined by the United Nations (UN). Thus, it was possible to verify, for example, that 49% of the articles studied the production of energy with wood waste, which contributes to advancing the goal concerning the value of the use of renewable sources; that (Figure 7) the most considered source of origin in the surveys was industrial waste, which helps to leverage the goals involving responsible industry and production; and that (Figure 9) the Global Warning Potential (GWP) impact category stands out as the most studied, which is fundamental to achieving the objective of controlling climate change. In this way, the results achieved in this research corroborate the Sustainable Development Goals (SDGs).

**4. Guideline Proposal**

The mapping performed in this study highlights the lack of standardization and clarity of the reviewed literature. For example, 67% did not clarify the type of analysis conducted (attributional or consequential), 31% omitted the type of software used, 27% did not explain the type of treatment of the residue, 8% did not explicate the functional unit or the limits

of the system studied, etc. These data show the need to standardize the application of LCA to environmentally analyze the elements for wood waste management scenarios, thus facilitating the interpretation and comparison of different studies.

From the identification of the deficiencies and good examples of the studies found in the reviewed literature, it was possible to elaborate a guideline proposal concerning the stages of the LCA methodology. As a recommendation, Figure 10 emphasizes the main points to be considered as a guideline for future LCA studies applied to the environmental assessment of wood waste management (WW) scenarios, thereby contributing to the development of better comparisons between the scenarios and improving the quality of work in this field.

Additionally, this study also has the potential to contribute to a possible draft standard to be developed specifically for LCA studies on the environmental performance of wood waste management systems.

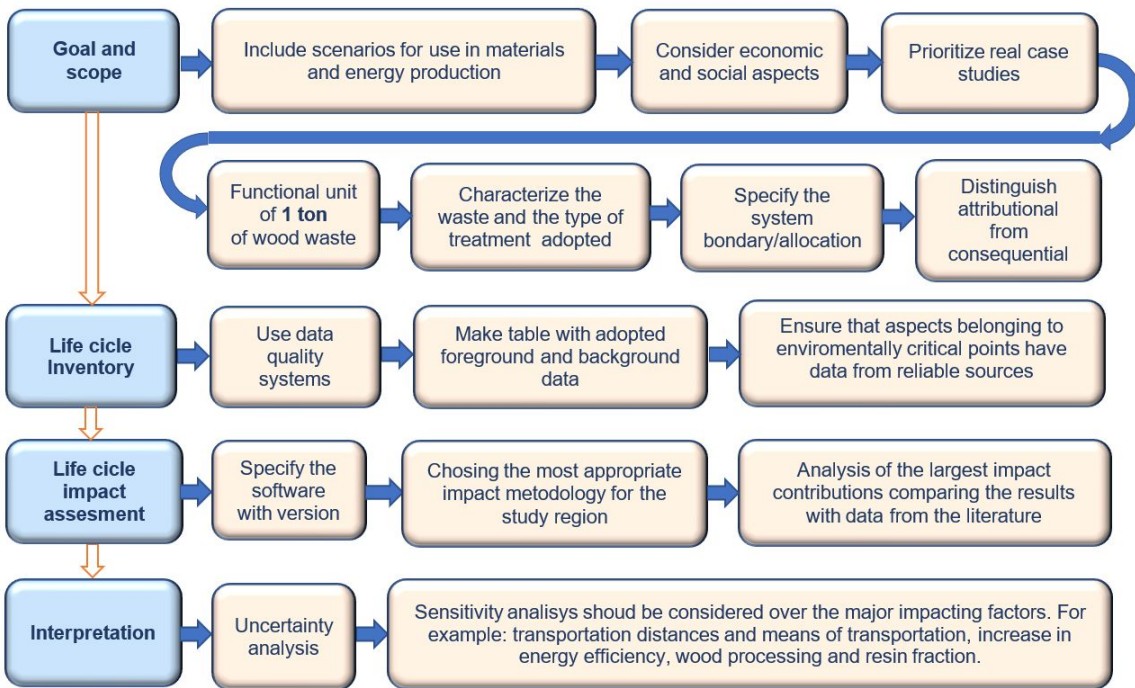

**Figure 10.** Guidelines for future LCA studies applied to the environmental assessment of WW management scenarios.

### *4.1. Consideration of the Guideline Proposal*

4.1.1. Definition of Objective and Scope

The comparison between potential scenarios that seek to study the use of waste material, for either energy production or materials, is important in order to realize better WW management solutions with different targets. In this review, most of the studies adopted real case studies, and this facilitates the collection of primary data. The functional unit should be suitable to comparisons between different applications; therefore, we recommend the use of 1 t of wood waste as a functional unit, since it proved to be adequate for all the objectives studied. Table 1 shows the main suggestions for future studies raised in the literature related to the five types of objectives classified in this review. Notably, all the objectives mention the need to analyze other impact categories besides climate change, perform sensitivity or uncertainty analyses, and include economic and social analyses. In this review, only 12 studies considered economic aspects, and none dealt with social aspects. In relation to the circular economy, the concept has been explored by some authors, but we observed that only eight studies explicitly mentioned the term "Circular Economy."

**Table 1.** Main suggestions for future studies regarding the types of objectives of the 49 articles.

| OBJECTIVE OF THE STUDY | EXISTING STUDIES | SUGGESTIONS |
|---|---|---|
| Cascade use of wood waste (CUWW) | [3] | Leverage waste in different scenarios based on quality |
| | [20] | Include waste treatment in the analysis |
| | [23] | Analyze resource efficiency against number of cascading steps |
| | [31] | Use thermo-hydrolytic disintegration method for the treatment of residue in the production of MDF and MDP (separate resin from wood). |
| | [31] | Deepen the studies of MDF and OSB |
| | [23,47] | Expand attributional to consequential LCA studies |
| | [48] | Analyze the extended combination of substitution of other materials and the cascading use of wood |
| Wood waste management strategies (WWMS) | [5] | Apply the concept of temporary storage of biomass carbon |
| | [14] | Leverage waste in different scenarios based on quality |
| | [49] | Include recycling, burning, and landfill in the same study |
| | [50] | Investigate the plastic wood production to replace virgin plastic |
| Virgin product X recycled product (VP × RP) recycled product (RP) | [6,28] | Investigate alternative resins as urea-formaldehyde replacement |
| | [10] | Investigate the production of cement panels with magnesium oxide cement (MOC) |
| Energy production using wood waste (EPWW) | [8] | Include a pre-treatment in the residue to remove the urea-formaldehyde resin |
| | [36] | Investigate effects of using low-cost additives in incineration plants that burn waste |
| | [44] | Evaluate the effect of biomass characteristics (moisture content and energy value) on the environmental performance of energy systems |

Note: Cascade use of wood waste [3,20,23,31,47,48]; Wood waste management strategies [5,14,49,50]; Virgin product X recycled product and recycled product [6,10,28]; Energy production using wood waste [8,36,44].

Studies generally focus on untreated waste, i.e., with properties similar to virgin wood, such as sawmill residues. Thus, the impacts of climate change related to the contaminants and chemical inputs of wood treatment are often disregarded from the consulted literature [8]. Most authors reported the type of treatment given to the residue, but few gave a detailed description of the process; only Taskhiri et al. (2019) [20] produced additional material on the subject.

The type of LCA conducted, either attributional or consequential, should be explained in the studies, since this aspect influences the entire modeling process performed. System boundaries should also be clearly defined, if possible, and shown in a detailed diagram form. In systems that encompass multiple products, allocation may be a necessary procedure. Based on a load-partitioning criterion, material, energy, and emissions flows should be allocated to the different products, and all considerations should be clearly documented and justified. Due to the possible influence of the analyst in the decisions made in this stage, ISO 14040 (ISO, 2006a) [18] encourages the use of process subdivision or system expansion to avoid allocation. The application of system expansion makes it difficult to conduct a study centered on only one product, which in some cases can lead to a less accurate study in relation to the individual life cycle of products. However, in studies in which a holistic view is part of the goal, such as cascading systems studies, this method is well-suited to analyzing combinations of life cycles [47].

### 4.1.2. Life Cycle Inventory

The life cycle inventory (LCI) phase necessitates a great deal of effort with respect to minimizing uncertainties and improving data quality. Thus, primary data should be adopted; if this proves to be impossible for all the included unitary processes, the complementary data should stem from peer-reviewed literature and commercial or free LCI databases. Databases are still incipient, and many are not suitable for certain geographical areas. In relation to the studied topic, in version 3.7 of Ecoinvent, we can find, for example, the inventory data for the production of MDP, in Brazil, provided in the study by Silva et al. (2013) [51]. In this sense, further research is needed to increase the availability of data, especially in developing countries.

This phase requires the use of data quality indicators such as the Pedigree Matrix, which allows for the transformation of a qualitative concept of data quality into quantitative values of variance from the following five indicators: reliability and completeness and temporal, geographical, and technological correlation [52,53].

The use of secondary data hampers the quantification of uncertainty, which could compromise the results of an LCA. However, recent studies point to better methods of adopting secondary data. Dai et al. (2022) [54] presented a very clear structure applicable to the realization of the LCI and the quantification of uncertainty. The developed structure has the potential to produce more reliable inventories for regions with smaller scales than those used in current databases. Zargar et al. (2022) [55] highlighted that, with regard to future contexts, existing LCI databases should be modified, such as the energy mix for the coming years. The authors also emphasized that collaboration between the LCA community and researchers in the area under study should be improved to ensure future enhancements.

For an understanding of the chosen processes and modifications made to the data from the literature or databases, all considerations should be clearly explained, and, if possible, gathered in a general summary table. Additionally, an effort must be applied to obtain data from trusted sources in all aspects. If this is not possible, the aspects of higher impact contributions should be prioritized. The main impact contributions highlighted in the research were process energy, transportation, and chemical inputs. In the latter, for example, the production of urea-formaldehyde resin, which is used in the manufacture of panels, presents relevant contributions for most categories of impact.

### 4.1.3. Life Cycle Impact Assessment

Several LCIA methods have been developed, as there is still no single methodology with which to associate LCI data with potential environmental impacts. According to the revised literature, in addition to the IPCC, methods with larger numbers of categories, such as ReCiPe, should be adopted for studies that opt for midpoint analyses [56]. For endpoint analyses, ReCiPe, Impact 2002+, and Eco-indicator were selected by the studied authors. Additionally, in the case of possible divergences in results, we advise the adoption of more than one impact method. Thus, a comparative analysis between the results will be possible.

The most-studied impact categories were global warming, acidification, eutrophication, human toxicity, ozone layer depletion, and ecotoxicity; these should be considered for further studies, depending on the objective to be achieved. Thus, the evaluation of the impact of the life cycle is the result of the chosen methodology, the analyzed categories of impact, and the software used. It is also emphasized that in order to guarantee the replicability of the studies, the specification of the LCIA software must be clarified.

### 4.1.4. Interpretation of the Results

The interpretation of the results must be in line with the objective presented in the study. Sensitivity and uncertainty analyses are required to increase research reliability. Uncertainty analysis was rarely performed in the reviewed studies, but sensitivity analyses were performed to reduce the uncertainties of the results. Only Faraca et al. (2019) [3] differed and performed a combined analysis of sensitivity and uncertainty, following the approach suggested by Bisinella et al. (2017) [57].

Some recurrent sensitivity analyses in the studies concerned the following aspects: distances and means of transport, energy source, increase in energy efficiency, wood processing, percentage of residue, and fraction of resins in the composition of agglomerated plates. These sensitivity analyses should be applied to verify the items with the most critical impact according to the results.

### 5. Conclusions and Prospects for Future Work

This article constitutes a comprehensive review of the literature—published from 2011 to 2021—related to LCA for the management of WW. The bibliometric results concerned the contemplation of the temporal evolution of the studies, the quantity of publications, the geographical distribution, and the journals that published the most studies in the field. The countries most present in the publications were Germany, Canada, and Italy when considering the main author. In 2019, the largest number of publications occurred, and the most relevant journal was the Journal of Cleaner Production.

The works selected to constitute the portfolio of this review were examined in detail according to ISO 14040 (2006a) [18] and ISO 14044 (2006b) [27]. In addition, through this mapping process, it was possible to verify that not all the studies presented the same degree of completeness in relation to the LCA methodology. Notably, the studies lacked transparency in relation to the information records and standardization of the work. It has also been emphasized that the results achieved in this research corroborate the United Nations Sustainable Development Goals (SDGs). Thus, the main conclusion of this study is that the authors need to adopt a global standardization of how to apply the LCA methodology in research that focuses on the environmental analysis of wood waste management systems.

Thus, in order for researchers to take a first step in this direction, to assist future studies of environmental performance, a guideline proposal (Figure 10) was presented, which shows the main points to be considered for the performance of better LCA. Although there is still a great deal of progress yet to be made, this initiative aims to contribute information such that new research in this area can be better compared between scenarios and improved in relation to the consistency of results. Additionally, in the near future, this article also has the potential to contribute to a possible draft standard to be developed specifically for LCA studies on the environmental performance of wood waste management systems.

From the analysis performed, several perspectives have been identified for future studies. The most relevant ones are summarized and highlighted below:

- According to the research focus, specific suggestions should be observed in future studies. Table 1 of item 4.1.1 highlights the main suggestions for future studies raised in the literature in relation to the five types of objectives classified in this review.
- The investigation of other categories of impact—in addition to climate change, the performance of sensitivity or uncertainty analyses, and the study of economic and social aspects—are relevant points for the expansion of LCA studies of WW.
- Within the research in this field, the concepts of the Circular Economy are still rarely explored. Studies with more comprehensive guidelines for waste management, which are oriented toward CE practices, should be encouraged.
- Few studies encompassed the environmental impacts related to the contaminants and chemical inputs of wood treatment. The research analyzed generally adopted untreated residues with properties similar to natural wood.
- For studies in which a holistic view is part of the objective, such as the studies on cascading systems, the application of system expansion approaches is more appropriate. Therefore, these approaches should be investigated in future studies.
- Current databases should be expanded so that all geographic areas can conduct reliable studies in their respective regions.
- There is still no single methodology for associating LCI data with potential environmental impacts. Therefore, it is important for research studies to adopt more than one method to obtain a more consistent analysis.

- Combined analyses of sensitivity and uncertainties are still rarely explored. All research papers should incorporate this type of analysis to reduce the uncertainties of the results.

**Author Contributions:** Conceptualization, G.C.d.S.P. and J.L.C.; methodology, G.C.d.S.P.; software, G.C.d.S.P.; validation G.C.d.S.P. and J.L.C.; formal analysis, G.C.d.S.P.; investigation G.C.d.S.P.; resources, J.L.C.; data curation G.C.d.S.P.; writing—original draft preparation, G.C.d.S.P.; writing—review and editing, G.C.d.S.P.; visualization, G.C.d.S.P. and J.L.C.; supervision, J.L.C. All authors have read and agreed to the published version of the manuscript.

**Funding:** This research was funded by Coordenação de Aperfeiçoamento de Pessoal de Nível Superior, Finance code 001, and Fundação de Amparo à Pesquisa e Inovação do Espírito Santo, project 107/2019.

**Institutional Review Board Statement:** The study did not require ethical approval.

**Informed Consent Statement:** Not applicable.

**Data Availability Statement:** Not applicable.

**Acknowledgments:** We are deeply grateful to Diego Lima Medeiros for his important considerations and fruitful discussions. We also appreciate the support granted by the Research Foundation of the State of Espírito Santo (Fundação de Amparo à Pesquisa e Inovação do Espírito Santo—FAPES).

**Conflicts of Interest:** The authors declare no conflict of interest.

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
