# Peer review of "LCA of Wood Waste Management Systems: Guiding Proposal for the Standardization of Studies Based on a Critical Review"

_sustainability, doi:10.3390/su15031854_

Round 1

Reviewer 1 Report

This study provides an interesting and timely review of the life cycle assessment approach to wood waste. The paper is well-written but can be revised to be more concise. Another issue is that the recommendation is not state-of-the-art and too general. I have the following comments and suggestions for the authors.

Line 64. I am not sure if the authors are still talking about the forestry sector or a broader sector. Please complete the sentence.

Line 71. I am not sure why the authors write this paragraph. It seems that this paragraph conveys the same information as the previous paragraph. Please revise.

Line 77. It is not clear which literature review the authors are talking about. Is it the cited literature or the authors’ work?

Line 168. The “01 publication” looks like a typo.

Line 233. It helps the readers if the authors could add the percentage after the “most research”.

Line 275. Need space at the beginning of the paragraph.

Figure 10. LCAs should include potential iterations so I would suggest adding order arrows among the blue boxes. I do not understand why “choosing the right software” is located in the LCIA part. If a LCA practitioner can access some kind of software, why won’t they just use it from the beginning and clarify that in the goal and scope step?

Table 1. I personally think this table is a bit confusing. I understand that the authors want to show how previous studies can be improved based on their framework but maybe it will be helpful to find another word to replace “recommendation”.

From Line 379 to the end of this section. I think these discussions are too general. Readers could potentially find these suggestions in textbooks. One way to improve may be to move out from wood waste and to introduce the ideas in the new methods and to include  more general discussion on the advancements in LCA (e.g., as summarized in Dai et al (2022). Environ. Sci. Technol. and Zargar et al. (2022)  Journal of Industrial Ecology).

Reviewer 2 Report

Dear Authors thank you so much for submission your manuscript in sustainability. The manuscript is up-to-date and the manuscript has clear novelty. The manuscript is well written and more importantly there is necessity for the standardization. However, there need to further improve the provided figures. I have noticed several mistakes on them and most of them are not clear, as example in Figure 3, the “other journal “ category was replaced by other papers.

I highly recommend the authors to link their findings and their proposal with the United Nations Sustainable Development Goals (SDGs), this will allow the different players to the adopted this proposal with other industry.

Thank you and good luck 

Round 2

Reviewer 1 Report

The authors have addressed all my comments. The last comment regarding the advancement in LCA has not been discussed thoroughly but considering its complexity and the lack of standardization, the author’s revision is acceptable.

This paper is ready for publication. I have the following minor suggestions for the authors, but I do not think the paper needs another iteration of review.

Line 22. I am curious about how many of the selected papers did not follow the ISO standardization. If the authors feel that the ISO standardization is not clear enough to guide existing studies on this specific topic, I recommend them to make stronger clarifications (e.g., what are the weaknesses of the existing standard, why it is hard to follow, and how to improve). The authors have provided a revised framework specified for wood waste. Are there suggestions the authors want to provide to the general LCA framework?  

Line 165: “Journals that have only 01 publication  …” should be “…1 publication…”.

Figure 3,4,6, 7,8 and 9. Please consider ranked the numbers and show the results in an ordered way.

Table 1. The column title “references” still looks strange. It may help if move the last column to the middle and show suggestions at the last column. The column titles could be 1) “Objective of the Study” or “Type of Study” 2) “Existing Studies” and 3) “Suggestions”. Another suggestion is the authors could rank the references and list them in an ordered fashion.

Line 333. Section 4 typo.
